# Learning to Represent Action Values as a Hypergraph on the Action Vertices

**Arash Tavakoli** [*]
Imperial College London

**Mehdi Fatemi**
Microsoft Research Montréal

**Petar Kormushev**
Imperial College London

## Abstract

Action-value estimation is a critical component of many reinforcement learning (RL) methods whereby sample complexity relies heavily on how fast a good estimator for action value can be learned. By viewing this problem through the lens of representation learning, good representations of both state and action can facilitate action-value estimation. While advances in deep learning have seamlessly driven progress in learning state representations, given the specificity of the notion of agency to RL, little attention has been paid to learning action representations. We conjecture that leveraging the combinatorial structure of multi-dimensional action spaces is a key ingredient for learning good representations of action. To test this, we set forth the *action hypergraph networks* framework—a class of functions for learning action representations in multi-dimensional discrete action spaces with a structural inductive bias. Using this framework we realise an agent class based on a combination with deep Q-networks, which we dub *hypergraph Q-networks*. We show the effectiveness of our approach on a myriad of domains: illustrative prediction problems under minimal confounding effects, Atari 2600 games, and discretised physical control benchmarks.

## 1 Introduction

Representation learning methods have helped shape recent progress in RL by enabling a capacity for learning good representations of state. This is in spite of the fact that, traditionally, representation learning was less often explored in the RL context. As such, the de facto representation learning techniques which are widely used in RL were developed under other machine learning paradigms (Bengio et al., 2013). Nevertheless, RL brings some unique problems to the topic of representation learning, with exciting headway being made in identifying and exploring such topics.

Action-value estimation is a critical component of the RL paradigm (Sutton & Barto, 2018). Hence, how to effectively learn estimators for action value from training samples is one of the major problems studied in RL. We set out to study this problem through the lens of representation learning, focusing particularly on learning representations of action in multi-dimensional discrete action spaces. While action values are conditioned on both state and action and as such good representations of both would be beneficial, there has been comparatively little research on learning action representations.

We frame this problem as learning a decomposition of the action-value function that is structured in such a way to leverage the combinatorial structure of multi-dimensional discrete action spaces. This structure is an inductive bias which we incorporate in the form of architectural assumptions. We present this approach as a framework to flexibly build architectures for learning representations of multi-dimensional discrete actions by leveraging various orders of their underlying sub-action combinations. Our architectures can be combined in succession with any other architecture for learning state representations and trained end-to-end using backpropagation, without imposing any change to the RL algorithm. We remark that designing representation learning methods by incorporating some form of structural inductive biases is highly common in deep learning, resulting in highly-publicised architectures such as convolutional, recurrent, and graph networks (Battaglia et al., 2018).

We first demonstrate the effectiveness of our approach in illustrative, structured prediction problems. Then, we argue for the ubiquity of similar structures and test our approach in standard RL problems.

---

[*]Correspondence to: Arash Tavakoli <a.tavakoli@imperial.ac.uk>.

Our results advocate for the general usefulness of leveraging the combinatorial structure of multi-dimensional discrete action spaces, especially in problems with larger action spaces.

## 2 BACKGROUND

### 2.1 REINFORCEMENT LEARNING

We consider the RL problem in which the interaction of an agent and the environment is modelled as a Markov decision process (MDP) $(\mathcal{S}, \mathcal{A}, P, R, S_0)$, where $\mathcal{S}$ denotes the state space, $\mathcal{A}$ the action space, $P$ the state-transition distribution, $R$ the reward distribution, and $S_0$ the initial-state distribution (Sutton & Barto, 2018). At each step $t$ the agent observes a state $s_t \in \mathcal{S}$ and produces an action $a_t \in \mathcal{A}$ drawn from its policy $\pi(. \mid s_t)$. The agent then transitions to and observes the next state $s_{t+1} \in \mathcal{S}$, drawn from $P(. \mid s_t, a_t)$, and receives a reward $r_{t+1}$, drawn from $R(. \mid s_t, a_t, s_{t+1})$.

The standard MDP formulation generally abstracts away the combination of sub-actions that are activated when an action $a_t$ is chosen. That is, if a problem has an $N^v$-dimensional action space, each action $a_t$ maps onto an $N^v$-tuple $(a_t^1, a_t^2, \ldots, a_t^{N^v})$, where each $a_t^i$ is a sub-action from the $i$th sub-action space. Therefore, the action space could have an underlying combinatorial structure where the set of actions is formed as a Cartesian product of the sub-action spaces. To make this explicit, we express the action space as $\mathcal{A} \doteq \mathcal{A}_1 \times \mathcal{A}_2 \times \cdots \times \mathcal{A}_{N^v}$, where each $\mathcal{A}_i$ is a finite set of sub-actions. Furthermore, we amend our notation for the actions $a_t$ into $\mathbf{a}_t$ (in bold) to reflect that actions are generally combinations of several sub-actions. Within our framework, we refer to each sub-action space $\mathcal{A}_i$ as an *action vertex*. As such, the cardinality of the set of action vertices is equal to the number of action dimensions $N^v$.

Given a policy $\pi$ that maps states onto distributions over the actions, the discounted sum of future rewards under $\pi$ is denoted by the random variable $Z_\pi(s, \mathbf{a}) = \sum_{t=0}^\infty \gamma^t r_{t+1}$, where $s_0 = s$, $\mathbf{a}_0 = \mathbf{a}$, $s_{t+1} \sim P(. \mid s_t, \mathbf{a}_t)$, $r_{t+1} \sim R(. \mid s_t, \mathbf{a}_t, s_{t+1})$, $\mathbf{a}_t \sim \pi(. \mid s_t)$, and $0 \le \gamma \le 1$ is a discount factor. The action-value function is defined as $Q_\pi(s, \mathbf{a}) = \mathbb{E}[Z_\pi(s, \mathbf{a})]$. Evaluating the action-value function $Q_\pi$ of a policy $\pi$ is referred to as a prediction problem. In a control problem the objective is to find an optimal policy $\pi^*$ which maximises the action-value function. The thesis of this paper applies to any method for prediction or control provided that they involve estimating an action-value function. A canonical example of such a method for control is Q-learning (Watkins, 1989; Watkins & Dayan, 1992) which iteratively improves an estimate $Q$ of the optimal action-value function $Q^*$ via

$$Q(s_t, \mathbf{a}_t) \leftarrow Q(s_t, \mathbf{a}_t) + \alpha \left( r_{t+1} + \gamma \max_{\mathbf{a}'} Q(s_{t+1}, \mathbf{a}') - Q(s_t, \mathbf{a}_t) \right), \tag{1}$$

where $0 \le \alpha \le 1$ is a learning rate. The action-value function is typically approximated using a parameterised function $Q_\theta$, where $\theta$ is a vector of parameters, and trained by minimising a sequence of squared temporal-difference errors

$$\delta_t^2 \doteq \left( r_{t+1} + \gamma \max_{\mathbf{a}'} Q_\theta(s_{t+1}, \mathbf{a}') - Q_\theta(s_t, \mathbf{a}_t) \right)^2 \tag{2}$$

over samples $(s_t, \mathbf{a}_t, r_{t+1}, s_{t+1})$. Deep Q-networks (DQN) (Mnih et al., 2015) combine Q-learning with deep neural networks to achieve human-level performance in Atari 2600.

### 2.2 DEFINITION OF HYPERGRAPH

A *hypergraph* (Berge, 1989) is a generalisation of a graph in which an edge, also known as a *hyperedge*, can join any number of vertices. Let $V = \{\mathcal{A}_1, \mathcal{A}_2, \ldots, \mathcal{A}_{N^v}\}$ be a finite set representing the set of action vertices $\mathcal{A}_i$. A hypergraph on $V$ is a family of subsets or hyperedges $H = \{E_1, E_2, \ldots, E_{N^e}\}$ such that

$$E_j \ne \emptyset \quad (j = 1, 2, \ldots, N^e), \tag{3}$$

$$\cup_{j=1}^{N^e} E_j = V. \tag{4}$$

According to Eq. (3), each hyperedge $E_j$ is a member of $\mathcal{E} = \mathcal{P}(V) \setminus \emptyset$, where $\mathcal{P}(V)$, called the *powerset* of $V$, is the set of possible subsets on $V$. The rank $r$ of a hypergraph is defined as the maximum cardinality of any of its hyperedges. We define a *c-hyperedge*, where $c \in \{1, 2, \ldots, N^v\}$,

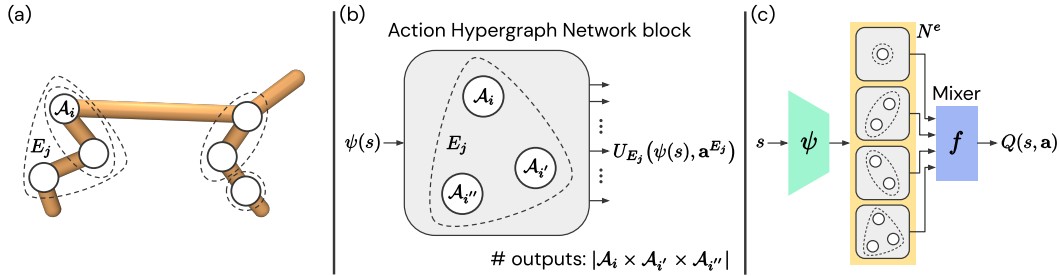

Figure 1: (a) A sample hypergraph overlaid on a physical system with six action vertices. (b) An instance building block of our framework for a sample hyperedge. (c) An architecture is realised by stacking several building blocks, one for each hyperedge in the hypergraph.

as a hyperedge with cardinality or order $c$. The number of possible $c$-hyperedges on $V$ is given by the binomial coefficient $\binom{N^v}{c}$. We define a $c$-*uniform* hypergraph as one with only $c$-hyperedges. As a special case, a $c$-*complete* hypergraph, denoted $K^c$, is one with all possible $c$-hyperedges.

## 3  ACTION HYPERGRAPH NETWORKS FRAMEWORK

We now describe our framework using the example in Fig. 1. Consider the sample physical system of Fig. 1a with six action vertices (solid circles). A sample hypergraph is depicted for this system with four hyperedges (dashed shapes), featuring a 1-hyperedge, two 2-hyperedges, and a 3-hyperedge. We remark that this set of hyperedges constitutes a hypergraph as there are no empty hyperedges (Eq. (3)) and the union of hyperedges spans the set of action vertices (Eq. (4)).

We wish to enable learning a representation of each hyperedge in an arbitrary hypergraph. To achieve this, we create a parameterised function $U_{E_j}$ (e.g. a neural network) for each hyperedge $E_j$ which receives a state representation $\psi(s)$ as input and returns as many values as the possible combinations of the sub-actions for the action vertices enclosed by its respective hyperedge. In other words, $U_{E_j}$ has as many outputs as the cardinality of a Cartesian product of the action vertices in $E_j$. Each such hyperedge-specific function $U_{E_j}$ is a building block of our *action hypergraph networks* framework.[1] Figure 1b depicts the block corresponding to the 3-hyperedge from the hypergraph of Fig. 1a. We remark that for any action $\mathbf{a} = (a^1, a^2, \ldots, a^{N^v})$ from the action space $\mathcal{A}$, each block has only one output that corresponds to $\mathbf{a}$ and, thus, contributes exactly one value as a representation of its respective hyperedge at the given action. This output $U_{E_j}(\psi(s), \mathbf{a}^{E_j})$ is identified by $\mathbf{a}^{E_j}$ which denotes the combination of sub-actions in $\mathbf{a}$ that correspond to the action vertices enclosed by $E_j$.

We can realise an architecture within our framework by composing several such building blocks, one for each hyperedge in a hypergraph of our choosing. Figure 1c shows an instance architecture corresponding to the sample hypergraph of Fig. 1a. The forward view through this architecture is as follows. A shared representation of the input state $s$ is fed into multiple blocks, each of which features a unique hyperedge. Then, a representation vector of size $N^e$ is obtained for each action $\mathbf{a}$, where $N^e$ is the number of hyperedges or blocks. These action-specific representations are then mixed (on an action-by-action basis) using a function $f$ (e.g. a fixed non-parametric function or a neural network). The output of this mixing function is our estimator for action value at the state-action pair $(s, \mathbf{a})$. Concretely,

$$Q(s, \mathbf{a}) \doteq f\left( U_{E_1}(\psi(s), \mathbf{a}^{E_1}), \, U_{E_2}(\psi(s), \mathbf{a}^{E_2}), \ldots, U_{E_{N^e}}(\psi(s), \mathbf{a}^{E_{N^e}}) \right). \tag{5}$$

While only a single output from each block is relevant for an action, the reverse is not the case. That is, an output from a block generally contributes to more than a single action's value estimate. In fact, the lower the cardinality of a hyperedge, the larger the number of actions' value estimates to which a block output contributes. This can be thought of as a form of combinatorial generalisation.

---

[1]Throughout this paper we assume linear units for the block outputs. However, other activation functions could be more useful depending on the choice of mixing function or the task.

Particularly, action value can be estimated for an insufficiently-explored or unexplored action by mixing the action's corresponding representations which have been trained as parts of other actions. Moreover, this structure enables a capacity for learning faster on lower-order hyperedges by receiving more updates and slower on higher-order ones by receiving less updates. This is a desirable intrinsic property as, ideally, we would wish to learn representations that exhaust their capacity for representing action values using lower-order hyperedges before they resort to higher-order ones.

## 3.1 MIXING FUNCTION SPECIFICATION

The mixing function receives as input a state-conditioned representation vector for each action. We view these action-specific representation vectors as *action representations*. Explicitly, these action representations are learned as a decomposition of the action-value function under the mixing function. Without a priori knowledge about its appropriate form, the mixing function should be learned by a universal function approximator. However, joint learning of the mixing function together with a good decomposition under its dynamic form could be challenging. Moreover, increasing the number of hyperedges expands the space of possible decompositions, thereby making it even harder to reach a good one. The latter is due to lack of identifiability in value decomposition in that there is not a unique decomposition to reach. Nevertheless, this issue is not unique to our setting as representations learned using neural networks are in general unidentifiable. We demonstrate the potential benefit of using a universal mixer (a neural network) in our illustrative bandit problems. However, to allow flexible experimentation without re-tuning standard agent implementations, in our RL benchmarks we choose to use summation as a non-parametric mixing function. This boils down the task of learning action representations to reaching a good linear decomposition of the action-value function under the summation mixer. We remark that the summation mixer is commonly used with value-decomposition methods in the context of cooperative multi-agent RL (Sunehag et al., 2017). In an informal evaluation in our RL benchmarks, we did not find any advantage for a universal mixer over the summation one. Nonetheless, this could be a matter of tuning the learning hyperparameters.

## 3.2 HYPERGRAPH SPECIFICATION

We now consider the question of how to specify a good hypergraph. There is actually not an all-encompassing answer to this question. For example, the choice of hypergraph could be treated as a way to incorporate a priori knowledge about a specific problem. Nonetheless, we can outline some general rules of thumb. In principle, including as many hyperedges as possible enables a richer capacity for discovering useful structures. Correspondingly, an ideal representation is one that returns neutral values (e.g. near-zero inputs to the summation mixer) for any hyperedge whose contribution is not necessary for accurate estimation of action values, provided that lower-order hyperedges are able to represent its contribution. However, as described in Sec. 3.1, having many mixing terms could complicate reaching a good decomposition due to lack of identifiability. Taking these into consideration, we frame hypergraph specification as choosing a rank $r$ whereby we specify the hypergraph that comprises all possible hyperedges of orders up to and including $r$:

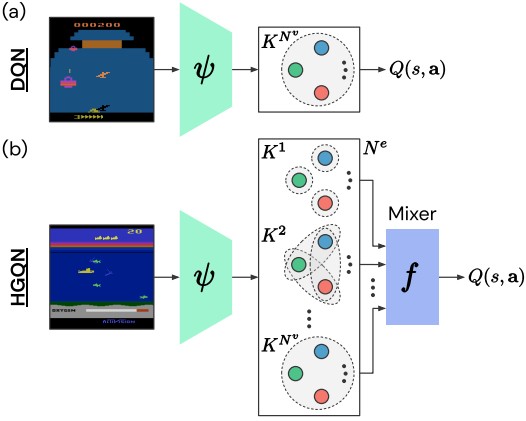

Figure 2: (a) A standard model. (b) A class of models in our framework, depicted by the ordered space of possible hyperedges $\mathcal{E}$. Our class of models subsumes the standard one as an instance.

$$H \doteq \cup_{c=1}^{r \leq N^v} K^c, \tag{6}$$

where hypergraph $H$ is specified as the union of $c$-complete hypergraphs $K^c$. Figure 2b depicts a class of models in our framework where the space of possible hyperedges $\mathcal{E}$ is ordered by cardinality.

On the whole, not including the highest-order ($N^v$) hyperedge limits the representational capacity of an action-value estimator. This could introduce statistical bias due to estimating $N$ actions' values

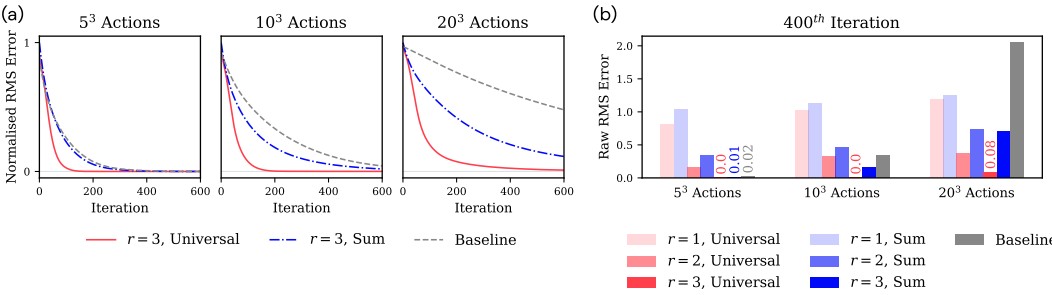

Figure 3: Prediction error in our illustrative multi-armed bandits with three action dimensions and increasing action-space sizes. Each variant is run on 64 reward functions in any action-space size. (a) Normalised average RMS error curves. (b) Average RMS errors at the 400th training iteration.

using a model with $M < N$ unique outputs.[2] In a structured problem $M < N$ could suffice, otherwise such bias is inevitable. Consequently, choosing any $r < N^v$ could affect the estimation accuracy in a prediction problem and cause sub-optimality in a control problem. Thus, preferably, we wish to use hypergraphs of rank $r = N^v$. We remark that this bias is additional to—and should be distinguished from—the bias of function approximation which also affects methods such as DQN. We can view this in terms of the bias-variance tradeoff with $r$ acting as a knob: lower $r$ means more bias but less variance, and vice versa. Notably, when the $N^v$-hyperedge is present even the simple summation mixer can be used without causing bias. However, when this is not the case, the choice of mixing function could significantly influence the extent of bias. In this work we generally fix the choice of mixing function to summation and instead try to include high-order hyperedges.

## 4 ILLUSTRATIVE PREDICTION PROBLEMS

We set out to illustrate the essence of problems in which we expect improvements by our approach. To do so, we minimise confounding effects in both our problem setting and learning method. Given that we are interested purely in studying the role of learning representations of action (and not of state), we consider a multi-armed bandit (one-state MDP) problem setting. We specify our bandits such that they have a combinatorial action space of three dimensions, where we vary the action-space sizes by choosing the number of sub-actions per action dimension from $\{5, 10, 20\}$.

The reward functions are deterministic but differ for each random seed. Despite using a different reward function in each independent trial, they are all generated to feature a good degree of decomposability with respect to the combinatorial structure of the action space. That is to say, by design, there is generally at least one possible decomposition that has non-zero values on all possible hyperedges. See Appendix A for details about how we generate such reward functions.

We train predictors for reward (equivalently, the optimal action values) using minimalist parameterised models that resemble tabular ones as closely as possible (e.g. our baseline corresponds to a standard tabular model) and train them using supervised learning. For our approach, we consider summation and universal mixers as well as increasingly more complete hypergraphs, specified using Eq. (6) by varying rank $r$ from 1 to 3. See Appendix A for additional experimental details.

Figure 3a shows normalised RMS prediction error curves (averaged over 64 reward functions) for two variants of our approach that leverage all possible hyperedges versus our tabular baseline. We see that the ordering of the curves remains consistent with the increasing number of actions, where the baseline is outperformed by our approach regardless of the mixing function. Nevertheless, our model with a universal mixer performs significantly better in terms of sample efficiency. Moreover, the performance gap becomes significantly wider as the action-space size increases, attesting to the utility of leveraging the combinatorial structure of actions for scaling to more complex action spaces. Figure 3b shows average RMS prediction errors at the 400th training iteration (as a proxy for "final"

---

[2]The notions of *statistical bias* and *inductive bias* are distinct from one another: an inductive bias does not necessarily imply a statistical bias. In fact, a good inductive bias enables a better generalisation capacity but causes little or no statistical bias. In this paper we use "bias" as a convenient shorthand for statistical bias.

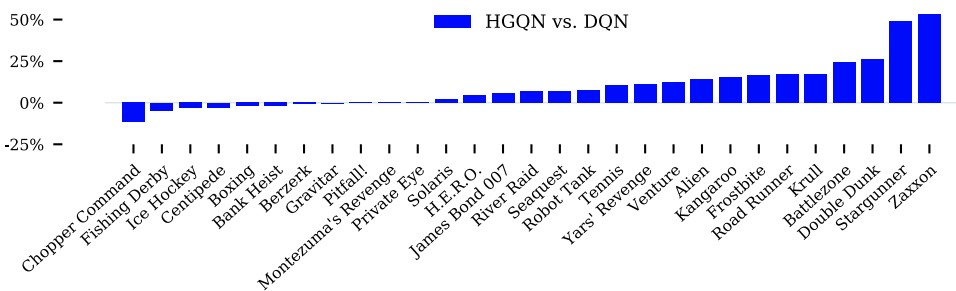

Figure 4: Difference in human-normalised score for 29 Atari 2600 games with 18 valid actions, HGQN versus DQN over 200 training iterations (positive % means HGQN outperforms DQN).

performance) for all variants in our study. As expected, including higher-order hyperedges and/or using a more generic mixing function improves final prediction accuracy in every case.

Going beyond these simple prediction tasks to control benchmarks in RL we anticipate a general advantage for our approach, following the same pattern of yielding more significant improvements in larger action spaces. Indeed, this is if similar structures are ubiquitous in practical RL tasks.

## 5 ATARI 2600 GAMES

We tested our approach in Atari 2600 games using the Arcade Learning Environment (ALE) (Bellemare et al., 2013). The action space of Atari 2600 is determined by a digital joystick with three degrees of freedom: three positions for each axis of the joystick, plus a button. This implies that these games can have a maximum of 18 discrete actions, with many not making full use of the joystick capacity. To focus our compute resources on games with a more complex action space, we limited our tests to 29 Atari 2600 games from the literature that feature 18 valid actions.

For the purpose of our control experiments, we combine our approach with deep Q-networks (DQN) (Mnih et al., 2015). The resulting agent, which we dub *hypergraph Q-networks* (HGQN), deploys an architecture based on our action hypergraph networks, similar to that shown in Fig. 2b, and using the summation mixer. Given that the action space has merely three dimensions, we instantiate our agent's model based on a hypergraph including the seven possible hyperedges. We realise this model by modifying the DQN's final hidden layer into a multi-head one, where each head implements a block from our framework for a respective hyperedge. To achieve a fair comparison with DQN, we ensure that our agent's model has roughly the same number of parameters as DQN by making the sum of the hidden units across its seven heads to match that of the final hidden layer in DQN. Specifically, we implemented HGQN by replacing the final hidden layer of 512 rectifier units in DQN with seven network heads, each with a single hidden layer of 74 rectifier units. Our agent is trained in the same way as DQN. Our tests are conducted on a stochastic version of Atari 2600 using *sticky actions* (Machado et al., 2018).

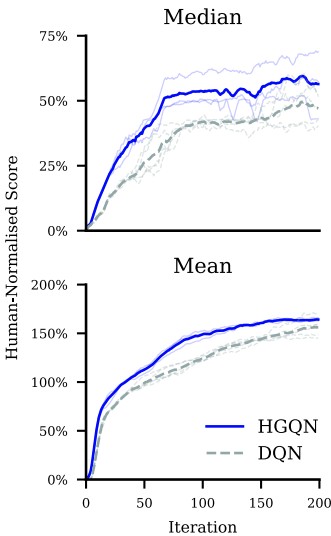

Figure 5: Human-normalised median and mean scores across 29 Atari 2600 games with 18 valid actions. Random seeds are shown as traces.

Figure 4 shows the relative human-normalised score of HGQN (three random seeds) versus DQN (five random seeds) for each game. Figure 5 shows median and mean human-normalised scores across the 29 Atari games for HGQN and DQN. Our results indicate improvements over DQN on the majority of the games (Fig. 4) as well as in overall performance (Fig. 5). Notably, these improvements are both in terms of sample complexity and final performance (Fig. 5). The consistency of improvements in these games has the promise of greater improvements in tasks with larger action spaces. See Appendix B for experimental details, including complete learning curves.

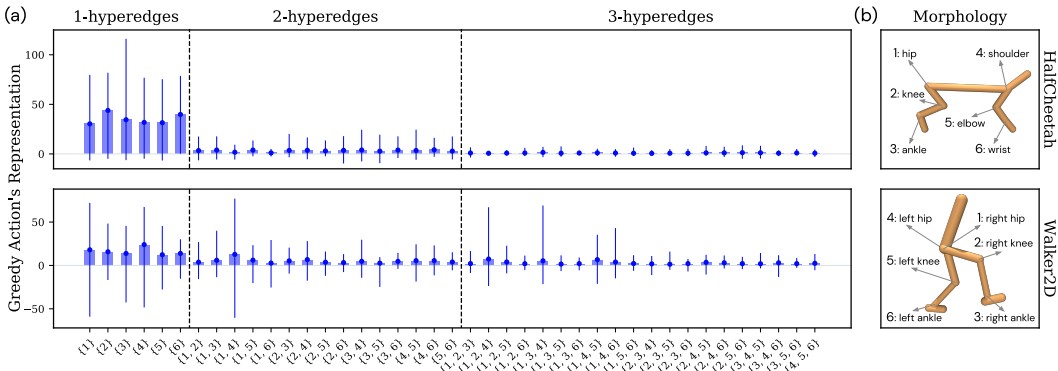

Figure 6: Learning curves for HGQN, DQN, and a simplified version of Rainbow in physical control benchmarks of PyBullet (nine random seeds). Shaded regions indicate standard deviation. Average performance of DDPG trained for 3 million environment steps is provided for illustration purposes.

Figure 7: (a) Average (bars) and minimum-maximum (error bars) per-hyperedge representation of the greedy action over 90000 steps (nine trained rank-3 HGQN models, 10000 steps each). (b) The actuation morphology of each respective domain is provided for reference.

To further demonstrate the versatility of our approach, we combine it with a simplified version of Rainbow (Hessel et al., 2018). We ran the resulting agent on the three best and worst-performing games from Fig. 4. We report our results in Appendix C. Notably, our results show that where HGQN outperforms DQN most significantly, its combination with the simplified version of Rainbow also outperforms the corresponding Rainbow baseline.

## 6 PHYSICAL CONTROL BENCHMARKS

To test our approach in problems with larger action spaces, we consider a suite of physical control benchmarks simulated using PyBullet (Coumans & Bai, 2019). These domains feature continuous action spaces of diverse dimensionality, allowing us to test on a range of combinatorial action spaces. We discretise each action space to obtain five sub-actions per joint. See Appendix D for experimental details, including network architecture changes for the non-pixel nature of states in these domains.

We ran HGQN with increasingly more complete hypergraphs, specified using Eq. (6) by varying rank $r$ from 1 to 3. Figure 6 shows the learning curves for HGQN and DQN. In low-dimensional Reacher and Hopper we do not see any significant difference. However, the higher-dimensional domains render DQN entirely ineffective. This demonstrates the utility of action representations learned by our approach. In Ant we only ran HGQN with a 1-complete model as the other variants imposed greater computational demands (see Sec. 8 for more information).

We provide the average final performance of DDPG (Lillicrap et al., 2016) to give a sense of the performances achievable by a continuous control method, particularly by one that is closely related to DQN. Interestingly, HGQN generally outperforms DDPG despite using the training routine of DQN, without involving policy gradients (Sutton et al., 1999). This may be due to local optimisation issues in DDPG which can lead to sub-optimal policies in multi-modal value landscapes (Metz et al., 2017; Tessler et al., 2019). Furthermore, we include the performance of DQN combined with

prioritised replay, dueling networks, multi-step learning, and Double Q-learning (denoted Rainbow[†]; see Hessel et al. (2018) for an overview). The significant gap between the performances of this agent and vanilla HGQN supports the orthogonality of our approach with respect to these extensions.

In Walker2D we see a clear advantage for including the 2-hyperedges, but additionally including the 3-hyperedges relatively degrades the performance. This suggests that a rank-2 hypergraph strikes the right balance between bias and variance, where including the 3-hyperedges causes higher variance and, potentially, overfitting. In HalfCheetah we see little difference across hypergraphs, despite it having the same action space as Walker2D. This suggests that the 1-complete model is sufficient in this case for learning a good action-value estimator.

Figure 7 shows average per-hyperedge representation of the greedy action learned by our approach. Specifically, we evaluated nine trained rank-3 HGQN models using a greedy policy for 10000 steps in each case. We collected the greedy action's corresponding representation at each step (i.e. one action-representation value for each hyperedge) and averaged them per hyperedge across steps. The error bars show the maxima and minima of these representations. In HalfCheetah the significant representations are on the 1-hyperedges. In Walker2D, while 1-hyperedges generally receive higher average representations, there is one 2-hyperedge that receives a comparatively significant average representation. The same 2-hyperedge also receives the highest variation of values across steps. This 2-hyperedge, denoted $\{1, 4\}$, corresponds to the left and right hips in the agent's morphology (Fig. 7b). A good bipedal walking behaviour, intuitively, relies on the hip joints acting in unison. Therefore, modelling their joint interaction explicitly enables a way of obtaining coordinated representations. Moreover, the significant representations on the 3-hyperedges correspond to those including the two hip joints. This perhaps suggests that the 2-hyperedge representing the hip joints is critical in achieving a good walking behaviour and any representations learned by the 3-hyperedges are only learned due to lack of identifiability in value decomposition (see Sec. 3).

## 7 RELATED WORK

Value decomposition has been studied in cooperative multi-agent RL under the paradigm of centralised training but decentralised execution. In this context, the aim is to decompose joint action values into agent-wise values such that acting greedily with respect to the local values yields the same joint actions as those that maximise the joint action values. A sufficient but not necessary condition for a decomposition that satisfies this property is the strict monotonicity of the mixing function (Rashid et al., 2018). An instance is VDN (Sunehag et al., 2017) which learns to decompose joint action values onto a sum of agent-wise values. QMIX (Rashid et al., 2018) advances this by learning a strictly increasing nonlinear mixer, with a further conditioning on the (global) state using hypernetworks (Ha et al., 2017). These are multi-agent counterparts of our 1-complete models combined with the respective mixing functions. To increase the representational capacity of these methods, higher-order interactions need to be represented. In a multi-agent RL context this means that fully-localised maximisation of joint action values is no longer possible. By relaxing this requirement and allowing some communication between the agents during execution, *coordination graphs* (Guestrin et al., 2002) provide a framework for expressing higher-order interactions in a multi-agent setting. Recent works have combined coordination graphs with neural networks and studied them in multi-agent one-shot games (Castellini et al., 2019) and multi-agent RL benchmarks (Böhmer et al., 2020). Our work repurposes coordination graphs from cooperative multi-agent RL as a method for action representation learning in standard RL.

Sharma et al. (2017) proposed a model on par with a 1-complete one in our framework and evaluated it in multiple Atari 2600 games. In contrast to HGQN, their model shows little improvement beyond DQN. This performance difference is likely due to not including higher-order hyperedges, which in turn limits the representational capacity of their model.

Using Q-learning in continuous action problems by discretisation has proven as a viable alternative to continuous control. Existing methods deal with the curse of dimensionality by factoring the action space and predicting action values sequentially (Metz et al., 2017) or independently (Tavakoli et al., 2018) across the action vertices. Another approach is to learn a (factored) proposal distribution for performing a sampling-based maximisation of action values (Van de Wiele et al., 2020). Our work introduces an alternative approach based on value decomposition and further enables a capacity for explicitly modelling higher-order interactions among the action vertices. We remark that our

1-complete models can achieve linear time complexity when used with a strictly monotonic mixer (e.g. summation) by enabling to find the action-value maximising actions in a decentralised manner.

Graph networks (Scarselli et al., 2009) have been combined with policy gradient methods for training modular policies that generalise to new agent's morphologies (Wang et al., 2018; Pathak et al., 2019; Huang et al., 2020). The global policy is expressed as a collection of graph policy networks that correspond to each of the agent's actuators, where every policy is only responsible for controlling its corresponding actuator and receives information from only its local sensors. The information is propagated based on some assumed graph structure among the policies before acting. Our work differs from this literature in many ways. Notably, our approach does not impose any assumptions on the state structure and, thus, is applicable to any problem with a multi-dimensional action space. The closest to our approach in this literature is the concurrent work of Kurin et al. (2021) in which they introduce a transformer-based approach to bypass having to assume a specific graph structure.

Chandak et al. (2019) explored learning action representations in policy gradient methods by a separate supervised learning process. Other methods have been proposed for learning in large discrete action spaces that have an associated underlying continuous action representation by using a continuous control method (van Hasselt & Wiering, 2009; Dulac-Arnold et al., 2015). Nevertheless, these methods do not leverage the combinatorial structure of multi-dimensional action spaces.

## 8 DISCUSSION AND FUTURE WORK

We demonstrated the potential benefit of using a more generic mixing function in our bandit problems (Sec. 4). However, in an informal study on a subset of our RL benchmarks, we did not see any improvements beyond our non-parametric summation mixer. Further exploration of more generic (even state-conditioned) mixing functions is an interesting direction for future work. In this case, comparisons with an action-in baseline architecture would be appropriate to investigate how much of the advantages of using a hypergraph model may come from sharing the parameters of mixing function across actions as opposed to modelling lower-order hyperedges.

The requirement to maximise over the set of possible actions limits the applicability of Q-learning in environments with high-dimensional action spaces. In the case of approximate Q-learning, this limitation is partly due to the cost of computing the possible actions' values before an exact maximisation can be performed. Our approach can bypass these issues in certain cases (e.g. when using a 1-complete hypergraph with a strictly monotonic mixer; see Sec. 7 for more information). To more generally address such issues, we can maximise over a sampled set of actions instead of the entire set of possible actions (Van de Wiele et al., 2020). An approximate maximisation as such will enable including higher-order hyperedges in environments with high-dimensional action spaces.

We demonstrated the practical feasibility of combining our approach with a simplified version of Rainbow. An extensive empirical study of the impact of these combinations as well as combining with further extensions, e.g. for distributional (Bellemare et al., 2017) and logarithmic RL (van Seijen et al., 2019), is left as future work. Moreover, a better understanding of how value decomposition affects the learning dynamics of approximate Q-learning could help establish which extensions are theoretically compatible with our approach.

Much could be done to improve learning in continuous action problems by discretisation. For instance, the performance of our approach could be improved by using exploratory policies that exploit the ordinality of the underlying continuous action structure instead of using $\varepsilon$-greedy. Moreover, DDPG uses an Ornstein-Uhlenbeck process (Uhlenbeck & Ornstein, 1930) to generate temporally-correlated action noise to achieve exploration efficiency in physical control problems with inertia. Using a similar inductive bias could further improve the performance of our approach in such problems. A more generally applicable approach which does not rely on the ordinality of the action space is to use temporally-extended $\varepsilon$-greedy exploration (Dabney et al., 2021). The latter could prove especially useful in discretised physical control problems as such. Another interesting opportunity is using a curriculum of progressively growing action spaces where a coarse discretisation can help exploration and a fine discretisation allows for a more optimal policy (Farquhar et al., 2020).

**Code availability** The source code can be accessed at: `https://github.com/atavakol/action-hypergraph-networks`.

ACKNOWLEDGEMENTS

We thank Ileana Becerra for insightful discussions, Nemanja Rakicevic and Fabio Pardo for comments on the manuscript, Fabio Pardo for providing the DDPG results which he used for benchmarks in Tonic (Pardo, 2020), and Roozbeh Tavakoli for help with visualisation. A.T. acknowledges financial support from the UK Engineering and Physical Sciences Research Council (EPSRC DTP). We also thank the scientific Python community for developing the core set of tools that enabled this work, including TensorFlow (Abadi et al., 2016) and NumPy (Harris et al., 2020).

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

## A  ILLUSTRATIVE PREDICTION PROBLEMS: EXPERIMENTAL DETAILS

**Environment**   We created each reward function by randomly initialising a hypergraph model with all possible hyperedges. Explicitly, in each independent trial, we sampled as many values as the total number of outputs across all possible hyperedges: sampling uniformly from $[-10, 10]$ for the 1-hyperedges, $[-5, 5]$ for the 2-hyperedges, and $[-2.5, 2.5]$ for the 3-hyperedges. This results in structured reward functions that can be decomposed to a good degree on the lower-order hyperedges but still need the highest-order hyperedge for a precise decomposition. Next, we generated a random mixing function by uniformly sampling the parameters of a single-hidden-layer neural network from $[-1, 1]$. The number of hidden units and the activation functions were sampled uniformly from $\{1, 2, \ldots, 5\}$ and {ReLU, tanh, sigmoid, linear}, respectively. Lastly, a deterministic reward for each action was generated by mixing the action's corresponding subset of values.

**Architecture**   We used minimalist parameterised models that resemble tabular ones as closely as possible. As such, our baseline corresponds to a standard tabular model in which each action value is estimated by a single unique parameter. In contrast, our approach does not require a unique parameter for each action as it relies on mixing multiple action-representation values to produce an action-value estimate. Therefore, for our approach we instantiated as many unique parameters as the total number of outputs from each model's hyperedges. For any model using a universal mixer, we additionally used a single hidden layer of 10 rectifier units for mixing the action-representation values, initialised using the Xavier uniform method (Glorot & Bengio, 2010).

**Training**   Each predictor was trained by backpropagation using supervised learning. We repeatedly sampled minibatches of 32 rewards (with replacement) to update a predictor's parameters, where each training iteration comprised 100 such updates. We used the Adam optimiser (Kingma & Ba, 2015) to minimise the mean-squared prediction error.

The number of hyperedges in our hypergraphs of interest (see Sec. 3.2) can be expressed in terms of binomial coefficients as

$$|H| \doteq \sum_{c=1}^{r \leq N^v} \binom{N^v}{c}, \tag{7}$$

where hypergraph $H$ is specified by its rank $r$ according to Eq. (6). As we discussed in Sec. 3, each action's value estimator is formed by mixing as many action-representation values as there are hyperedges in the model (Eq. (5)). In our simple models for the bandit problems, each such value corresponds to a single parameter. As such, each learning update per action involves updating as many parameters as the number of hyperedges. Therefore, to ensure fair comparisons, we adapted the learning rates across different models in our study based on the number of parameters involved in updating each action's value estimate. Concretely, we set a single effective learning rate across all models in our study. We then obtained each model's individual learning rate $\alpha$ via

$$\alpha \doteq \frac{\text{effective learning rate}}{|H|}. \tag{8}$$

We used an effective learning rate of 0.0007 in our study. While this achieves the same actual effective learning rate for both the baseline and our models using the summation mixer, the same does not hold exactly for our models using a universal mixer. Nonetheless, this still serves as a useful heuristic to obtain similar effective learning rates across all models. Importantly, the baseline receives the largest individual learning rate across all other models—especially one that improved its performance with respect to any other learning rates used by the variants of our approach.

The logic behind the adjustments of learning rate here simply does not hold in the case of high-capacity, nonlinear models such as those used in our RL benchmarks. Therefore, in our RL benchmarks we used the same learning rate across all variants.

**Evaluation**   The learning curves were generated by calculating the RMS prediction error across all possible actions after each training iteration per independent trial.

## B    ATARI 2600 GAMES: EXPERIMENTAL DETAILS

We based our implementation of HGQN on the Dopamine framework (Castro et al., 2018). Dopamine provides a reliable open-source code for DQN as well as enables standardised benchmarking in the ALE under the best known evaluation practices (see, e.g., Bellemare et al. (2013); Machado et al. (2018)). As such, we conducted our Atari 2600 experiments without any modifications to the agent or the environment parameters with respect to those outlined in Castro et al. (2018), except for the network architecture change for HGQN (see Sec. 5). Our DQN results are based on the published Dopamine baselines. We limited our tests in Atari 2600 to all the games from Wang et al. (2016) that feature 18 valid actions excluding Defender for which DQN results were not provided as part of the Dopamine baselines.

The human-normalised scores reported in this paper were given by the formula (similar to van Hasselt et al. (2016); Dabney et al. (2018))

$$\frac{\text{score}_{\text{agent}} - \text{score}_{\text{random}}}{\text{score}_{\text{human}} - \text{score}_{\text{random}}}, \tag{9}$$

where $\text{score}_{\text{agent}}$, $\text{score}_{\text{human}}$, and $\text{score}_{\text{random}}$ are the per-game scores (undiscounted returns) for the given agent, a reference human player, and random agent baseline. We used Table 2 from Wang et al. (2016) to retrieve the human player and random agent scores.

The relative human-normalised score of HGQN versus DQN in each game (Fig. 4) was given by the formula (similar to Wang et al. (2016))

$$\frac{\text{score}_{\text{HGQN}} - \text{score}_{\text{DQN}}}{\max(\text{score}_{\text{DQN}}, \text{score}_{\text{human}}) - \text{score}_{\text{random}}}, \tag{10}$$

where $\text{score}_{\text{HGQN}}$ and $\text{score}_{\text{DQN}}$ were computed by averaging over their respective learning curves.

Figure 8 shows complete learning curves across the 29 Atari 2600 games with 18 valid actions.

## C    ATARI 2600 GAMES: ADDITIONAL RESULTS

We combined our approach with a simplified version of Rainbow (Hessel et al., 2018) that includes prioritised replay (Schaul et al., 2016), dueling networks (Wang et al., 2016), multi-step learning (Hessel et al., 2018), and Double Q-learning (van Hasselt et al., 2016). We did not include C51 (Bellemare et al., 2017) as it is not trivial how it can effectively be combined with our approach. We also did not combine with noisy networks (Fortunato et al., 2018) which are not implemented in Dopamine. We denote the simplified Rainbow by Rainbow[†] and the version combined with our approach by HG-Rainbow[†].

We ran these agents on the three best and worst-performing games from Fig. 4. We conjecture that the games in which HGQN outperforms DQN most significantly should feature a kind of structure that is exploitable by our approach. Therefore, given that our approach is notionally orthogonal to the extensions in Rainbow[†], we can also expect to see improvements by HG-Rainbow[†] over Rainbow[†]. Figure 9 shows the relative human-normalised score of HG-Rainbow[†] versus Rainbow[†] (three random seeds in each case) along with those of HGQN versus DQN from Fig. 4 for these six games, sorted according to Fig. 4 (blue bars). We see that, generally, the signs of relative scores for Rainbow[†]-based runs are aligned with those for DQN-based runs. Remarkably, in Stargunner the relative improvements are on par in both cases.

This experiment mainly serves to demonstrate the practical feasibility of combining our approach with several DQN extensions. However, as each extension in Rainbow[†] impacts the learning process in certain ways, we believe that extensive work is required to establish whether such extensions are theoretically sound when combined with the learning dynamics of value decomposition. In fact, a recent study on the properties of linear value-decomposition methods in multi-agent RL could hint at a potential theoretical incompatibility with certain replay schemes (Wang et al., 2020). Therefore, we defer such a study to future work.

Figure 10 shows complete learning curves across the six select Atari 2600 games.

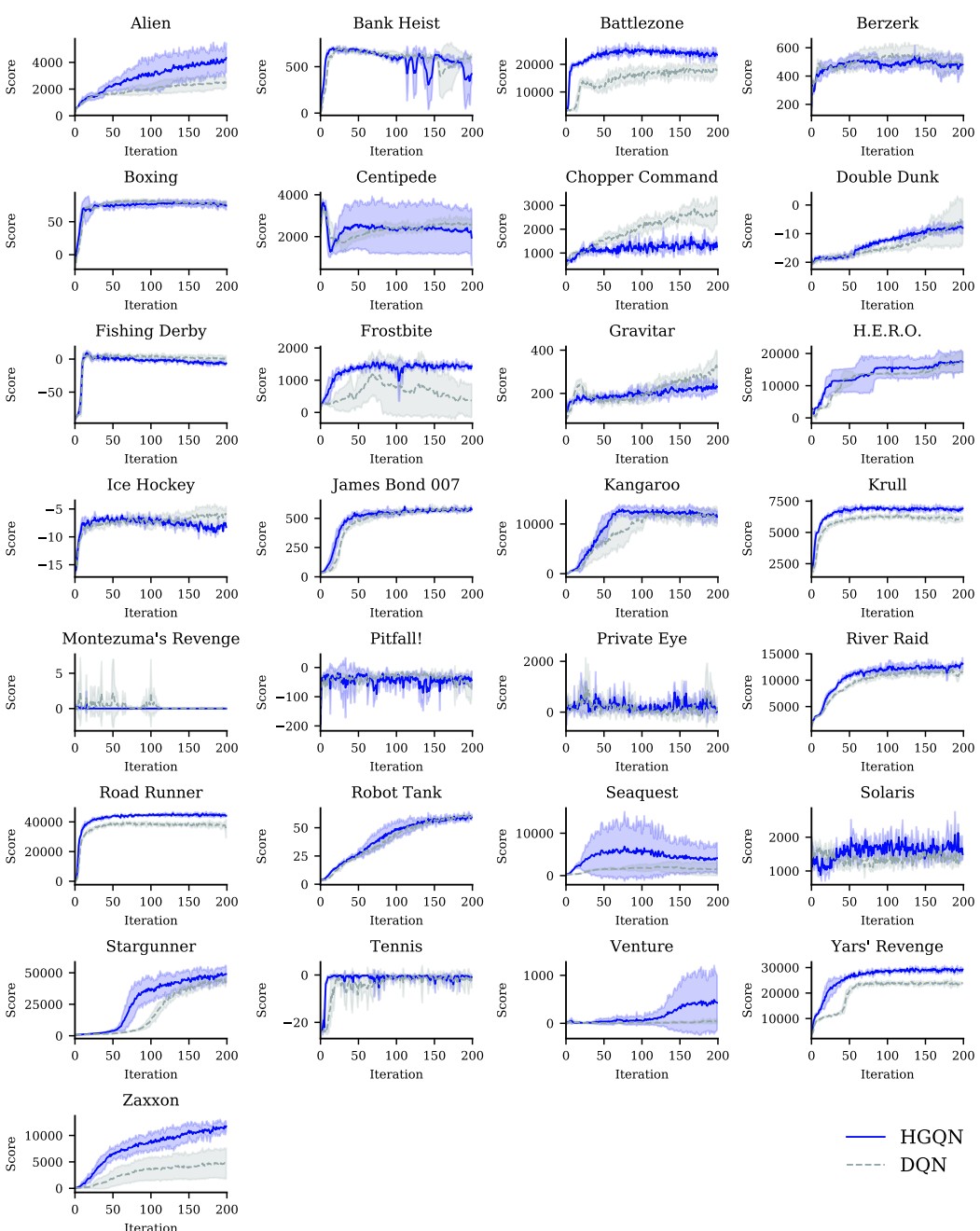

Figure 8: Learning curves in 29 Atari 2600 games with 18 valid actions for HGQN and DQN. Shaded regions indicate standard deviation.

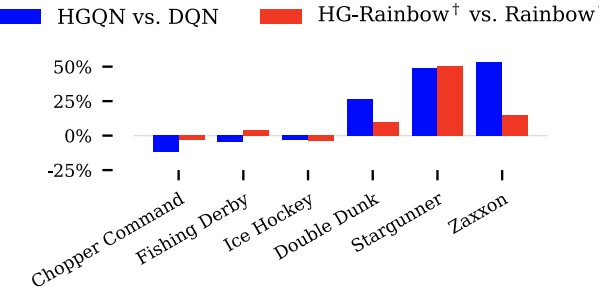

Figure 9: Difference in human-normalised score for six Atari 2600 games with 18 valid actions, featuring the three best and worst-performing games from Fig. 4 (positive % means HGQN outperforms DQN or HG-Rainbow[†] outperforms Rainbow[†] over 200 training iterations).

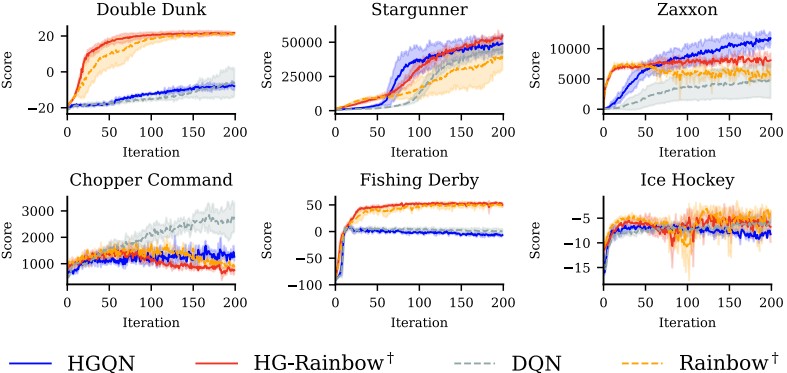

Figure 10: Learning curves for HGQN, HG-Rainbow[†], DQN, and Rainbow[†] in six Atari 2600 games with 18 valid actions, featuring the three best and worst-performing games from Fig. 4.

## D    PHYSICAL CONTROL BENCHMARKS: EXPERIMENTAL DETAILS

**Environment**    We tested our approach on a suite of physical control benchmarks simulated using PyBullet (Coumans & Bai, 2019). PyBullet benchmarks provide a free and open-source alternative to the standard MuJoCo (Todorov et al., 2012) benchmarks of OpenAI Gym (Brockman et al., 2016). The five environments in our experiment feature all PyBullet benchmarks excluding those with one-dimensional action spaces (i.e. InvertedPendulum and InvertedDoublePendulum) or without a stable implementation (i.e. Humanoid). Table 1 shows dimensionality and size of the action spaces in these environments, where the latter is obtained by discretising each action space with the granularity of five sub-actions per joint. By default, these environments use predefined time limits: 1000 steps for all locomotion tasks and 150 steps for the single manipulation task (i.e. Reacher).

**Architecture**    We simplified the network architectures by replacing the convolutional networks with two hidden layers of 600 and 400 rectifier units to reflect the non-pixel nature of states. Moreover, we adapted the final hidden layer from 512 to 400 rectifier units, applying the same heuristic of dividing them equally across the number of network heads for HGQN as in our Atari 2600 experiments. We remark that this heuristic does not achieve similar numbers of parameters with respect to our baselines in environments with large action spaces. Nevertheless, this is inevitable given that such standard models require a set of unique parameters for representing each action's value.

**Training**    We left the rewards unchanged, unlike in our Atari 2600 experiments where the rewards were clipped to deal with the great variations of the scale of rewards from game to game.

We generally used the same agent implementations as in our Atari 2600 experiments, with the exception of the following changes. Bootstrapping from non-terminal timeout transitions can significantly improve performance in environments which have short auxiliary time limits (Pardo et al., 2018). As such, in our physical control environments which have relatively short time limits, we applied this technique to all agents in our experiment (including DDPG). This is especially admissible in our case given that all agents in our study are off-policy. We outline any other changes to the learning hyperparameters with respect to those used in our Atari 2600 experiments in Table 2 (see Extended Data Table 1 of Mnih et al. (2015) for descriptions of hyperparameters).

**Evaluation**    The learning curves (Fig. 6) were generated by evaluating the agents every 10000 steps during the training, each time for a minimum of 5000 steps (until the last evaluation episode ends by reaching a terminal state or the time limit).

**Continuous control baseline**    The reported DDPG results are based on Spinning Up in Deep RL (Achiam, 2018) which provides a reliable open-source code for DDPG. The final performances (at 3 million environment steps) were obtained by averaging test performances (with no action noise) over the last 100000 steps of training (20 random seeds). The network architecture and hyperparameters match those reported in Lillicrap et al. (2016).

## E    PHYSICAL CONTROL BENCHMARKS: SUPPLEMENTAL VIDEOS

In Table 3 we provide links to supplemental videos showing representative behaviours learned by HGQN in a subset of PyBullet benchmarks together with the representation of the chosen action at each step. The videos are created using an $\varepsilon$-greedy policy with $\varepsilon = 0.001$ on models that were trained for 3 million environment steps.

Table 1: Action spaces in our physical control benchmarks.

| DOMAIN | DIMENSIONALITY | SIZE |
|---|---|---|
| Reacher | 2 | 25 |
| Hopper | 3 | 125 |
| HalfCheetah | 6 | 15625 |
| Walker2D | 6 | 15625 |
| Ant | 8 | 390625 |

Table 2: Hyperparameters used for HGQN and DQN in our physical control benchmarks.

| HYPERPARAMETER | VALUE |
|---|---|
| minibatch size | 64 |
| replay memory size | 100000 |
| agent history length | 1 |
| target network update frequency | 2000 |
| action repeat | 1 |
| update frequency | 1 |
| optimiser | Adam (Kingma & Ba, 2015) |
| learning rate | 0.00001 |
| $\hat{\epsilon}$ (a constant used for numerical stability in Adam) | 0.0003125 |
| $\beta_1$ ($1^{\text{st}}$ moment decay rate used by Adam) | 0.9 |
| $\beta_2$ ($2^{\text{nd}}$ moment decay rate used by Adam) | 0.999 |
| loss function | mean-squared error |
| initial exploration | 1 |
| final exploration | 0.05 |
| final exploration step | 50000 |
| replay start size | 10000 |

Table 3: Supplemental videos of HGQN in a subset of our physical control benchmarks.

| DOMAIN | VIDEO URL |
|---|---|
| Hopper | https://youtu.be/aOhqikV0gVA |
| HalfCheetah | https://youtu.be/2-f63SGyUHM |
| Walker2D | https://youtu.be/N3DQrN90h7U |

