# OpenReview forum: "Learning to Represent Action Values as a Hypergraph on the Action Vertices"
_ICLR.cc/2021/Conference — ICLR 2021 Poster_

### Official Review · AnonReviewer3 · 2020-10-26
**Good paper but some clarifications are needed!**

**Rating:** 8
**Confidence:** 4

**Review:**

The paper considers the problem of representation learning of actions, i.e., learning a decomposition of action-value function in multidimensional action spaces using hypergraphs. The key idea is to represent the actions as a hypergraph and learn a representation for each hyperedge in an arbitrary hypergraph. Consequently, the architecture conduits of a neural network for each hyperedge that are then combined using a mixing function. The choice of mixing function appears to be dependent on the problem -- for instance, it could be an universal mixer such as a neural  network or it could be a simple summation. The key impact of this architecture could be in the bandit problem setting with very large action spaces. Empirical results on a few atari games (29) and simulated physical control benchmarks clearly demonstrate the superiority of the approach.

The paper is written well. The problem is well motivated and sufficient details are provided. Experimental results appear convincing. The use of hypergraphs themselves are not necessarily novel but the implications in large multi-agent systems makes the paper compelling. Personally, i feel that the paper can have a good impact.

I do have a few comments:

First, I think the term “relational inductive bias” is quite a stretch for here. It really is much more than a simple inductive bias. You are constructing the full architecture. Granted that the structure serves as a bias but this is much more than an inductive bias. It really can be interpreted as a search bias as well.
Moreover, relational is also misused a lot (not just in this paper). IF you claim relational, then you should demonstrate generalization. Without that one cannot claim relational. I dont think your method is generalizable to varying number of objects in the domain. I suggest calling these as structured problems more than relational domains. The atari games for instance are not even close to being relational.
I am not sure I understand why a 3-hyperedge will have higher variance. Could you kindly explain this? Does this mean that you are overfitting in terms of creating newer edges that dont exist? Is it too fine-grained of a representation? Some speculation could certainly improve the paper.

---

> ### Author Response · Authors · 2020-11-18
> **Thank you for taking an interest in our paper and for your thoughtful feedback.**
>
> Thank you for taking an interest in our paper and for your thoughtful feedback.
>
> * We agree that using the term "*relational* inductive bias" is perhaps a bit of a stretch given that our approach, in its current form, does not generalise to new sub-action spaces. In the revised version, we have replaced this term with "*structural* inductive bias" which is more accurate as you correctly pointed out. Nonetheless, we believe that the structure incorporated in our architectures is an inductive bias, in the same way that Convolutional, Recurrent, and Graph Nets are architectures with different inductive biases (as discussed in Battaglia et al. (2018)).
>
> * Generally, when a hyperedge is included that is in fact not needed for achieving a sufficient level of accuracy in a certain task, then it increases the representational capacity of the model unnecessarily for that task. This added representational capacity causes higher variance (as the search space for a good function increases). As you correctly pointed out, such a model is more subject to overfitting. For instance, for Walker2D in Fig. 6 we see that going from 1-hyperedges to additionally including 2-hyperedges improves performance significantly. But when we additionally add the 3-hyperedges performance degrades.
> When we inspect the learned representations in our 3rd-order model in Fig. 7, we see that considerable values are learned on some of the 3-hyperedges, while we know that if these values were zero (similar to our 2nd-order model) a better behaviour could be achieved. This can be interpreted as a form of overfitting which occurs because value-decomposition is non-identifiable (i.e. there is not a unique decomposition, and the space of possible decompositions increases with increasing number of hyperedges). (We have refined our explanation of this in paragraph 3 of Sec. 6 to clarify this.)

---

### Official Review · AnonReviewer2 · 2020-10-28
**Considering a representation for action space (on top of state space) is a good idea but technics are incremental.**

**Rating:** 6
**Confidence:** 2

**Review:**

This work considers the idea of adding a representation for action space. This is on top of the usual representation of state-space. Their approach is based on hypergraph representation and shows its merits in the experimental results. But in general, I find the technical contributions (borderline) incremental.

---

> ### Author Response · Authors · 2020-11-18
> **Thanks for taking the time to read the paper.**
>
> Thanks for taking the time to read the paper.

---

### Official Review · AnonReviewer1 · 2020-10-29
**A novel and general approach to decomposing action-value function**

**Rating:** 8
**Confidence:** 4

**Review:**

This work focuses on learning action representations for problems involving high-dimensional action spaces. The aim is to build a flexible and general methodology for learning representations of multidimensional actions that can be combined with existing architectures (which mostly focus on learning state representations). The ideas initially suggested in Sharma et al (2017) have been generalized and refined in this work leading to impressive empirical results.

Strengths:
* The hypergraph formulation of the action space is well motivated and well grounded in literature.
* The aims set out in the introduction of the paper were duly justified -- the proposed framework is indeed very flexible and general as can be seen from the application to multiple problem types.
* The choice for Mixing function and Hypergraph specification are practical which makes them broadly applicable.
* The empirical results clearly illustrate the benefits achieved due to action representation.
* The paper is clearly written and well-organized.

Weaknesses:
* The experiments with three dimensional actions (Sec 4, 5) demonstrate clear benefits of the method, however, the action dimensionality is small enough to include all possible hyperedges. Sec 6 conducts experiments using five-dimensional actions where we notice tradeoffs between using hypergraphs of different ranks. From Fig 6 & 7, it is unclear what are the benefits of using hyperedges of higher order.
* It would be informative to conduct experiments on problems with higher dimensional action spaces such as Hunters & Rabbits.
* The experiments lack comparison against other methods that decompose actions such as Metz et al (2017).

Update after author response:
I thank the authors for their detailed rebuttals as well as for engaging with the reviewers. It'd be worth adding a small section on how to scale this method for higher dimensional action spaces (bullet 2 in author response). Comparison with Metz et al (2017) is valid and appreciated. Based on this, I am happy to increase my rating from 7 to 8.

---

> ### Author Response · Authors · 2020-11-18
> **Thank you for taking an interest in our paper and for your thoughtful feedback.**
>
> Thank you for taking an interest in our paper and for your thoughtful feedback.
>
> * Just to clarify, HalfCheetah and Walker2D have 6-dimensional actions and Ant has 8-dimensional actions (see Table 1). In Walker2D and Hopper (Fig. 6) we certainly see a clear advantage for going beyond 1-hyperedges, where additionally including the 2-hyperedges improved learning performance significantly. One finding of our study in Sec. 6 is that these standard physical control benchmarks are simple enough to be handled rather well even by only using the 1-hyperedges. Nevertheless, many practical problems of interest can be imagined, such as recommender systems (similar to our bandit problems in Sec. 4), for which including the higher-order hyperedges would be highly beneficial. We'd like to think of this using the analogy of Taylor series, where oftentimes expanding only the first few terms is sufficient in gaining a good approximation but there will always be settings where much higher-order terms would be necessary to get a good enough approximation.
> * Testing in domains with higher-dimensional actions such as the suggested Hunters & Rabbits is definitely interesting. Nevertheless, scaling our approach to higher-dimensional actions requires either combining our approach with a different algorithm than Q-learning (e.g. using our approach as the critic in an Actor-Critic method) or using approximate maximisation in Q-learning so that we do not need to evaluate all actions (see paragraphs 1-2 of Sec. 8 for more information). Currently, the only variant of our approach that scales gracefully to much higher-dimensional domains (with linear-time complexity as opposed to exponential-time in standard DQN) is the 1-complete model combined with a monotonic mixer (e.g. our summation mixer), which we have tested in Ant with an 8-dimensional action space (the highest-dimensional task in the PyBullet suite with a reliable environment implementation). Therefore, we consider testing in higher-dimensional domains as a direction for future work.
> * Metz et al. (2017) considered the atomic factorisation of the action space (comparable to a 1-complete model in our framework) where actions are sequentially selected for each action dimension. This reduces the class of possible Q-functions that can be represented (i.e. the ordering generally introduces bias). In our framework, we deal with sets (without ordering) which is in this sense more general. Nevertheless, the ordering should likely not matter much in our physical control benchmarks and, as such, we agree that it would be interesting to see how SDQN fairs against our approach. Unfortunately, no (stable) open-source implementation of this method is available, which makes reproducing their results difficult.

---

### Official Review · AnonReviewer4 · 2020-10-29
**Kind of novel but not sure about scalability and how significant improvement is**

**Rating:** 5
**Confidence:** 2

**Review:**

This paper incorporates a concept called hypergraph network into reinforcement learning. The idea of hypergraph is to extend edge to hyperedge where a set of vertices can be considered at the same time. This seems natural for scenarios like continuous action control with multi-dimension action space. From experimental results this proposed action hypergraph networks outperform several existing baselines.

Pros:
This paper has relatively high novelty. The idea of hypergraph may be useful for broad range of applications.

Cons:
1. HGQN vs. DQN is this a good comparison? DQN is a good baseline but no longer SOTA for Atari games. I wonder how HGQN vs. rainbow or DDPG for Figure 4 looks like.
2. It will be better if authors provide error bands for Figure 5. It's unclear how significant the proposed method is better.
3. When number of actions gets large, this hypergraph idea will have exponential complexity. How to find a best way to partition the graph (or generalize to 3+ hyperedges)?

---

> ### Author Response · Authors · 2020-11-18
> **Thank you for your time and thoughtful feedback.**
>
> Thank you for your time and thoughtful feedback.
>
> #### Q1:
> * Figure 8 shows the difference in human-normalised score for six Atari games using Rainbow$^{\dagger}$ (see the last paragraph in Sec. 5 for more information).
> * DDPG relies on differentiating the Q-function which is only possible in continuous-action spaces (and so it is not possible to test DDPG in Atari).
> * Figure 6 shows that HGQN (our model combined with DQN) outperforms DDPG in almost all domains and also significantly outperforms Rainbow$^{\dagger}$ in physical control problems (see paragraph 2 in Sec. 6 for more information).
>
> #### Q2:
> We have now updated Fig. 5 to additionally show individual random seeds as traces.
>
> #### Q3:
> The issue of exponential-complexity in number of action dimensions is due to the maximisation operation in Q-learning (because we have to evaluate all actions' values to be able to perform a maximisation over them). As such, this issue also applies to standard models such as DQN (i.e. this issue is not unique to our models). In fact, our approach enables handling this issue much more efficiently under certain conditions. For instance, a model with only 1-hyperedges and a monotonic mixer (e.g. our summation mixer) can achieve *linear-time complexity* in action dimensionality as opposed to *exponential-time complexity* in a standard model such as DQN. (See paragraphs 1-2 of Sec. 7 for more information.)
>
> From the standpoint of the increased number of model parameters needed to represent more hyperedges, we should remark that our framework enables having significantly less number of parameters: In DQN the number of parameters in the last layer *grows exponentially* in action dimensions, while we can achieve a significantly lower number of parameters by using a model with only lower-order hyperedges. For example, in an 8-dimensional task with 5 sub-actions per joint (such as our Ant environment), DQN needs $5^8=390625$ outputs, while a complete 4th-order model in our approach has $8 \times 5 + 28 \times 5^2 + 56 \times 5^3 + 70 \times 5^4 = 51490$ (i.e. a standard model needs nearly 8 times more outputs; which results in even a much larger number of parameters). From this viewpoint, a standard model like DQN generally scales much less gracefully in comparison to a model in our framework that does not include the highest-order hyperedge.
>
> Now, how to enable higher-order hyperedges (e.g. beyond 3) more efficiently is a distinct matter. One important aspect is to enable sparse hypergraphs (e.g. not having to instantiate all possible c-hyperedges). One way is to find a good hypergraph model using architecture search in a meta-learning context. Another way is to adaptively prune hyperedges during the course of training based on, e.g., some measure of uncertainty in estimates of each hyperedge. We have added a short paragraph at the end of Sec. 8 to outline learning of sparse hypergraphs (or partitions) as a direction for future work.

---

### Official Review · AnonReviewer5 · 2020-11-04
**Interesting paper that raises interesting questions.**

**Rating:** 7
**Confidence:** 5

**Review:**

EDIT: The authors clarified the presentation, gave a nuanced response to my concerns about the relative scalability, and promised to discuss the relationship to the actions-in architecture in the final version (it's not as central as I first thought given the limited role played by universal mixers). Solid work in its current state.

The authors propose a method that incorporates multi-way relationships between action dimensions in the estimate of action-value functions. This provides a strong inductive bias for reinforcement learning problems with a small number of action dimensions, each  taking several values. This is an interesting approach, but there are presentation and experimental issues preventing me from endorsing its publication.

Currently the presentation obscures some of the limitations of the method. Not until the experiments is it revealed that the method (in its current state) is only applicable to tasks with finite action spaces. Most tasks typically considered as having multiple action dimensions are continuous control tasks, so this is unexpected given the motivation. Not knowing this up front also hurts the clarity of the paper e.g. Figure 1b mentions outputting all possible actions, which is only coherent in the context of this finite cardinality restriction. Now, I'm not saying that not handling continuous action spaces is necessary (your results on discretized continuous action tasks are impressive), just that you should be more up front with this fact.

A second hidden limitation has to do with your method's scaling. If I understand your method correctly, to compute the maximum action-value you have execute the hyper-graph and mixer repeatedly, once for each possible action. This is in stark contrast to the standard actions-out architecture popularized by DQN -- a single forward pass compute all action-values. You (correctly) point out that both your method and DQN grow exponentially in the number of action dimensions, but in practice the latter will scale much better as it just amounts to a larger dimension matmul. Again, this limitation need not be overcome, but it should be mentioned.

On the experimental front, there is a serious baseline missing. Prior to DQN, many RL (with function approximation) papers treated the action as an input. Indeed, the only reason this is less common these days is due to the run-time issue discussed above. Treating the action as an input corresponds to only using the highest order hyper-graph (since the action contains all action dimensions), making that special case not a contribution of this paper. Thus, comparing against this case would be necessary to show that this method improves upon prior work. Additionally comparing against more recent approaches (e.g. auto-regressive processing of action dimensions) would further strengthen the paper, but this is of less importance in my opinion.

---

> ### Author Response · Authors · 2020-11-18
> **Thank you for your time and thoughtful feedback.**
>
> Thank you for your time and thoughtful feedback.
>
> #### Response to paragraph 2:
> In the revised version, we have now clarified early on (in the abstract and introduction) that our approach currently applies to *discrete* (finite) action spaces and, as such, our physical control problems are discretised. Specifically, we state multi-dimensional *discrete* action spaces and *discretised* physical control problems to emphasise this.
>
> #### Response to paragraph 3:
> To maximise the Q-values in Q-learning, we do a forward pass to retrieve the values of all hyperedges *once*. Then, we pass these values through the mixer on an action-by-action basis (i.e. as you correctly pointed out). Our implementation performs the latter in batched form to be more efficient. However, as you have correctly pointed out, this still has a higher run-time than a single pass in, e.g., DQN; that is **assuming the same number of model parameters** for both DQN and HGQN. (Our discussion in paragraph 3 of Sec. 3 should now be clear about this.)
>
> Nevertheless, we disagree with your general statement that "DQN will scale much better" in terms of run-time solely based on this. Our approach offers computational efficiency elsewhere which can be much more substantial in comparison:
> * Our framework enables having significantly less number of model parameters: In DQN the number of parameters in the last layer *grows exponentially* in action dimensions, while we can achieve a significantly lower number of parameters by using a model with only lower-order hyperedges. For example, by using a model with only 1-hyperedges, the number of parameters *grows linearly* in action dimensions. (This relates to our discussion in the last paragraph of Sec. 3.2, where a lower-order approximation allows for a smaller number of parameters.)
> * By considering a model with only 1-hyperedges and a monotonic mixer (e.g. our summation mixer), we can find the action that maximises the Q-values by maximising locally on the 1-hyperedges (i.e. without having to pass through the mixer). This achieves *linear-time complexity* in action dimensionality as opposed to *exponential-time complexity* in a standard model such as DQN. (See paragraphs 1-2 of Sec. 7 for more information.)
>
> Moreover, the above discussions are specific to the combination of our approach with Q-learning. However, our approach is not specific to Q-learning and can be combined with other action-value methods. For instance, our approach can be used to learn a critic in an *actor-critic method* where an action drawn from the actor only needs to be evaluated by the critic in a *single forward pass* through the model (similar to a standard model; since only one action's value needs to be evaluated). Interestingly, this also allows us to have a continuous-action actor used with a discrete-action critic. (We have now included this discussion in paragraph 2 of Sec. 8, just after we discuss scaling issues due to maximisation in Q-learning.)
>
> #### Response to paragraph 4:
> I. We respectfully disagree with your statement that a DQN-variant with action-in-input is a serious/more relevant baseline in our study than the standard DQN with all-actions in output for the following reason:
>
> * In Fig. 2, consider using a HGQN model with only the highest-order hyperedge and a summation mixer. This precisely yields DQN; because the summation mixer only receives as input one value per action and thus has no effect on the inputs (no mixing takes place). As such, the standard DQN model is actually the most relevant baseline to compare against for HGQN (which also uses a summation mixer). Had we experimented with a neural network mixing function in our HGQN models, we would then agree that a baseline with only the highest-order hyperedge and a mixing function is appropriate. Lastly, as we have stated in the caption of Fig. 2, our framework is a generalisation of the standard model where we additionally enable lower-order hyperedges (i.e. only using the highest-order hyperedge is indeed not a contribution of our paper).
>
> II. Thank you for the suggestion to test against auto-regressive processing of action dimensions. In the Related Work section, we have discussed Metz et al. (2017) which considered the atomic factorisation of the action space where sub-actions are sequentially selected for each action dimension. This reduces the class of possible Q-functions that can be represented (i.e. the ordering generally introduces bias). In our framework, we deal with sets (without ordering) which is in this sense more general. Nevertheless, we agree that it would be interesting to compare against this method. Unfortunately, no (stable) open-source implementation of this method is available, which makes reproducing their results difficult.

---

> > ### Comment · AnonReviewer5 · 2020-11-24
> > **Re: Missing Baseline**
> >
> > "I. We respectfully disagree with your statement that a DQN-variant with action-in-input is a serious/more relevant baseline in our study than the standard DQN with all-actions in output for the following reason:
> >
> > In Fig. 2, consider using a HGQN model with only the highest-order hyperedge and a summation mixer. This precisely yields DQN; because the summation mixer only receives as input one value per action and thus has no effect on the inputs (no mixing takes place). As such, the standard DQN model is actually the most relevant baseline to compare against for HGQN (which also uses a summation mixer). Had we experimented with a neural network mixing function in our HGQN models, we would then agree that a baseline with only the highest-order hyperedge and a mixing function is appropriate. Lastly, as we have stated in the caption of Fig. 2, our framework is a generalisation of the standard model where we additionally enable lower-order hyperedges (i.e. only using the highest-order hyperedge is indeed not a contribution of our paper)."
> >
> > I agree that your method 1) can't represent actions-in DQN without a neural network mixer and 2) generalizes by allowing lower-order hyperedges. However, I must stress that "highest-order hyperedge with a mixing function" is known prior work (i.e. the default prior to DQN), though obviously not known by that name. As such, it is needed as a baseline to establish how your summation mixer comparing to existing methods. And while your lower-order results don't face this criticism, your highest-order results (Atari) constitute is significant enough fraction of the overall contribution to warrent this baseline's inclusion.

---

> > > ### Author Response · Authors · 2020-11-24
> > > **Thanks for your response.**
> > >
> > > We believe that an actions-in Q-network in Atari will have lower performance than standard DQN for the following reason:
> > > * **Interference due to parameter-sharing:** Providing action in input corresponds to parameter-sharing across the actions, which subjects the learning updates to interference. Whether one should embrace parameter-sharing is a distinct inductive bias which should be studied by itself. However, it is reasonable to assume that in small action spaces (such as Atari's with 18 actions) using such a parameter-sharing scheme will generally hurt performance.
> > > It is important to note that our HGQN models also *do not* perform parameter-sharing on the hyperedge-networks or the mixing function (which is a non-parametric summation).
> > >
> > > Let's draw a parallel to help clarify our point:
> > > In a task with very small state-action spaces, one would not wish to use function approximation over tabular learning (because the latter can be trained much more stably and with higher learning rates due to *no interference* in updates). Going from the tabular-style output of standard DQN (i.e. one output per possible action) in Atari to parameter-sharing across actions introduces additional approximation and, thus, interference where such approximation (w.r.t. actions) is truly not needed, given how small the action space is (only 18 actions).
> > >
> > > Lastly, comparing against an actions-in variant of DQN takes enormous amounts of time and resources since it would require hyperparameter tuning to achieve a fair comparison (e.g. excessive number of shared-parameters complicates the learning process, while small number of shared-parameters introduces higher approximation-error/bias). This goes beyond the scope of this paper while due to the interference caused by parameter-sharing, we expect standard DQN to give superior performance to its actions-in variant.

---

> > > > ### Comment · AnonReviewer5 · 2020-11-24
> > > > **An interesting hypothesis worth testing.**
> > > >
> > > > Yes, there could be interference due to parameter-sharing, but that is exactly the sort of empirical hypothesis that I believe is worth investigating.
> > > >
> > > > "Going from the tabular-style output of standard DQN (i.e. one output per possible action) in Atari to parameter-sharing across actions introduces additional approximation and, thus, interference where such approximation (w.r.t. actions) is truly not needed, given how small the action space is (only 18 actions)."
> > > >
> > > > A very similar argument could be made for your own approach -- its not obvious that exploiting higher-order structure would yield benefits on such a small action space. But your method does indeed show benefits, and by similar reasoning its not a priori obvious that this baseline wouldn't also.

---

> > > > > ### Author Response · Authors · 2020-11-24
> > > > > **Thanks for your reply.**
> > > > >
> > > > > This empirical hypothesis is worth investigating, but nevertheless it is an orthogonal hypothesis to our approach and, as such, requires a separate paper to study it in depth.
> > > > >
> > > > > Let's clarify why we believe this to be an orthogonal hypothesis:
> > > > >
> > > > > Our results in Atari exploit lower-order hyperedges in addition to the highest-order hyperedge which is used by standard models (DQN) in order to test the hypothesis that lower-order hyperedges help to learn more efficiently. Note that here **neither the baseline DQN nor our HGQN agents use an action-in style of models**; since we use a non-parametric mixing function in HGQN (as discussed in our earlier response and already agreed by the reviewer).
> > > > > The hypothesis that the reviewer is suggesting to investigate relates to the question: *whether using parameter-sharing across the actions a useful inductive bias in Atari games despite the added interference in updates?* This is a distinct question which makes investigating this hypothesis out of the scope of our paper (especially given the enormous amounts of time and resources it needs to do hyperparameter tuning to achieve a fair comparison.)

---

> > > > > > ### Comment · AnonReviewer5 · 2020-11-24
> > > > > > **A reasonable compromise?**
> > > > > >
> > > > > > The "actions-in" architecture has existed in the literature for a long time and it is equivalent to using the highest order hyper-edge with a universal mixer. As such, it is prior work that your approach subsumes. Generally a paper is strengthened by showing such special cases don't capture all of the advantages of the full approach. Without showing this, I find it unclear whether or not multiple hyper-edges are advantageous when combines with a universal mixer.
> > > > > >
> > > > > > However, it is true that you don't use the universal mixer in your major results, so this would add some tuning burden to do so solely for the purpose of this comparison. As such, I'd be happy with this baseline being mentioned in the paper as future work.

---

> > > > > > > ### Author Response · Authors · 2020-11-24
> > > > > > > **Thanks!**
> > > > > > >
> > > > > > > This is a good compromise: we will add a paragraph about this in the *future work* section. However, given that the open discussion phase is fast coming to an end, we may not be able to add this paragraph in a revised version before the deadline. We hope that you accept for this minor addition to appear in the camera-ready version.

---

> > > > > > > > ### Comment · AnonReviewer5 · 2020-11-24
> > > > > > > > **Agreed :)**
> > > > > > > >
> > > > > > > > Sure thing, I have faith that you'll update it for camera-ready -- I'll update my review now.

---

> > > > > > > > > ### Author Response · Authors · 2020-11-24
> > > > > > > > > **Thanks :)**
> > > > > > > > >
> > > > > > > > > Thank you for all the feedback and the positive review!

---

### Decision · Program_Chairs · 2021-01-07
**Final Decision**

**Decision:**

Accept (Poster)

**Comment:**

After reading the reviews and rebuttal and looking over the paper, I feel that the results are indeed strong, and the paper could have an impact in terms of exploiting the relationship among action dimensions. Maybe the only detail that I would add is that going through the example given in Fig 1 completely could be useful as it might not be perfectly obvious how (e.g. considering a simple mixing function like summation) one retrieves the q values for someone not familiar with this particular topic.